# Measuring threshold and latency of motion perception on a swinging bed

**Maxime Guyon**[1], **Cyrielle Chea**[1], **Davy Laroche**[2,3], **Isabelle Fournel**[4], **Audrey Baudet**[2], **Michel Toupet**[1,5], **Alexis Bozorg Grayeli**[1,6]*

**1** Otolaryngology Department, Dijon University Hospital, Dijon, France, **2** INSERM CIC 1432, Plateforme d'Investigation Technologique, Dijon University Hospital, Dijon, France, **3** INSERM UMR1093, Cognition, Action et Plasticité Sensorimotrice, Université de Bourgogne Franche Comté, Dijon, France, **4** INSERM CIC 1432, Module Epidémiologie Clinique/Essais Cliniques, Dijon, France, **5** Centre d'Explorations Fonctionnelles Otoneurologiques, Paris, France, **6** CNRS UMR 6306, Le2i Research Laboratory, Dijon, France

\* alexis.bozorggrayeli@chu-dijon.fr

## Abstract

### Introduction

Our objective was to develop and to evaluate a system to measure latency and threshold of pendular motion perception based on a swinging bed.

### Materials and methods

This prospective study included 30 healthy adults (age: 32 ± 12 years). All subjects were tested twice with a 10 min. interval. A second trial was conducted 2 to 15 days after. A rehabilitation swinging bed was connected to an electronic device emitting a beep at the beginning of each oscillation phase with an adjustable time lag. Subjects were blindfolded and auditory cues other than the beep were minimized. The acceleration threshold was measured by letting the bed oscillate freely until a natural break and asking the patient when he did not perceive any motion. The perception latency was determined by asking the patient to indicate whether the beep and the peak of each oscillation were synchronous. The time lag between sound and peak of the head position was swept from -750 to +750 ms by 50 ms increments.

### Results

The mean acceleration threshold was 9.2±4.60 cm/s². The range width of the synchronous perception interval was estimated as 535±190 ms. The point of subjective synchronicity defined as the center of this interval was -195±106 ms (n = 30). The test-retest evaluation in the same trial showed an acceptable reproducibility for the acceleration threshold and good to excellent for all parameters related to sound-movement latency.

### Conclusion

Swinging bed combined to sound stimulation can provide reproducible information on movement perception in a simple and non-invasive manner with highly reproducible results.

**Data Availability Statement:** All relevant data are within the paper and its Supporting Information files.

**Funding:** This study was supported by the Société ORL de Bourgogne and the Centre Hospitalier Universitaire de Dijon (to MG).

**Competing interests:** The authors have declared that no competing interests exist.

**Abbreviations:** GVS, galvanic stimulation; ICC, interclass correlation; PS, peak-sound interval; PSS, point of subjective synchronicity; RSD, relative standard deviation; SD, standard deviation; SP, sound-peak interval; SPI, synchronous perception interval.

# Introduction

Today, there is no routine test to evaluate the vestibular input by the awareness of the body movements. Previous attempts to measure psycho vestibular parameters such as the perception threshold of body acceleration are based on complex and expensive systems which cannot be easily applied to all dizzy fragile or old subjects [1]. To evaluate the perception of circular movements in healthy subjects, Nooij et al. employed a MPI Cybermotion Simulator [2]. Sensitivity to vertical self-motion was evaluated in healthy volunteers on a similar device by Nesti et al. [3]. Other authors set up a Moog motion platform to detect dynamic tilt thresholds in patients with vestibular migraine [4] or a motor-driven linear sled on a 4.2-m track to assess linear movement perception [1]. The complexity of the setups, the duration of the examination, their cost and cumbersomeness hamper their clinical use in routine. In this view, a pendular movement on a rehabilitation swinging bed appears as a more accessible and probably a safer approach to the exploration of movement perception. Indeed, to our knowledge, none of these experimental platforms comply to the safety requirements for a routine clinical use in contrast to physiotherapy swinging beds.

Evaluating the movement perception has major potential applications. Falls in senior subjects have a major medico-economic impact and their prevention is a significant challenge for many health actors [5, 6]. The pathophysiology of balance disorders in the elderly is complex and probably variable from one patient to another [7]. Disturbances in functional connectivity, slower central processing and reaction to the disequilibrium are significant mechanisms in senior fallers among several others such as lower weight of vestibular input, lack of coordination, sarcopenia and inadequate reaction [7–10]. More generally, the awareness of the body movement is a prerequisite to adapted postural reactions and to rehabilitation in subjects with balance disorders. A decline of this capacity is observed with sedentary lifestyle and age [11]. This awareness can be characterized by several parameters (e.g., change of direction relative to the gravity vector, relative movement of body parts, change of location in space) among which, the perception threshold of body acceleration and the delay of this perception. Indeed, the impaired perception of fall timing appears to be related to the risk of fall in the elderly [12].

Measuring the delay of body movement perception is distorted by the delay in the subject's response if it is manual or vocal. Hence, perception delays of different sensory modalities are compared to each other. Previous works have already demonstrated that this type of comparison for vestibular, visual and auditory stimuli provide consistent results in terms of processing delay at a conscious level [13]. These studies have shown that visual and auditory inputs are processed more rapidly than vestibular information [14].

Multisensory integration of visual, vestibular, proprioceptive, and auditory cues for movement perception is crucial in balance and seems to be affected by diseases such as vestibular migraine [15] or age [16]. We hypothesized that this integration could be assessed by exploring the synchronous perception of a sound and a passive body oscillation on a swinging bed. Measuring acceleration perception threshold has potential implications on understanding the mechanisms of dizziness and fall [17]. Threshold values are subject to significant variation depending on the plane of the stimulation and stimulus profile (sinus, linear, steps, etc.) [1]. We hypothesized that we could measure a reproducible threshold on the swinging bed during deceleration. From a practical standpoint, measuring 2 potentially important parameters (synchronous perception of sound and movement and acceleration perception threshold) on the same device and with the same setup would be interesting in a clinical setup.

The aim of this study was to develop a system to measure the delays for which sound and body movement were perceived as synchronous, and the threshold of acceleration perception

on a safe device applicable to clinical routine and to evaluate its tolerance and reliability in healthy adults.

## Materials and methods

This monocentric pilot study was conducted on 30 healthy young adults in a tertiary referral center for balance disorders. We estimated the population size, by setting $\alpha = 0.05$, $\beta = 0.1$, the value of Cronbach's alpha at null hypothesis = 0, and the expected value of Cronbach's alpha = 0.7. The population size was estimated at 24 according to Bujang et al. [18] and increased to 30 to account for potential lost to follow-up at the retest. The population included 16 men and 14 women with a mean age of 32 years (range: 20–61). Subjects with past medical history of balance disorders or hearing disabilities were excluded. The protocol was reviewed and approved by the institution's ethical committee (CPP Est III) and a written consent was obtained from all subjects. We have complied with APA ethical standards in the treatment of the subjects.

A total of 4 tests was designed for each subject and each parameter. After inclusion, subjects underwent a trial of test and retest measuring the latency and the acceleration threshold of movement perception on a swinging bed. A 10-minute interval separated the test and the retest. A second test-retest trial was carried out several days after the first (mean delay between trials 13± 2.1 days, range: 2–50), on the same group. Four subjects were lost to follow-up for the second trial.

### Experimental set-up

Subjects were placed on a swinging bed suspended to a 2.5 m-high gantry (Fig 1). Sound and friction were minimized by ball-bearings on the rotation axis. The radius of the oscillation was 2.4 m. Preliminary tests showed a 1% variation of this radius as a function of the weight of the subject. The swinging movement was initiated by a manual backward traction of the bed and a silent release. For the measurement of acceleration threshold perception, the amplitude of this initial displacement was controlled by a laser beam projected on a scale (millimetric resolution) on the ground. To measure the latency of the movement perception, an infrared detector was placed on the ground to detect the passage of the bed at its lowest point at each cycle. This device was connected to a processor and a loudspeaker inside the detector box and approximately 1.5 meters from the subject's ears enabling the system to produce a beep (5 ms, 80 dB SPL) at the beginning of each oscillation (subject's head at its highest position, peak). Considering the speed of sound (343 m/s), this distance created a 4 ms delay. During the first 3 bed passages in front of the infrared detector (half cycles), the device measured and averaged the half cycles of the oscillation. Then, the system began to emit a beep with a negative or a positive time lag based on this calculated period. The oscillation period of this compound pendulum is stable for small oscillations (1–2 rad) as in our case. In this way, the position of the head could be estimated and anticipated with precision. The delay between the peak and the beep could be adjusted by the operator with 50 ms increments.

### Measurement protocol

The subject was installed on the bed on his/her back comfortably and blinded by a mask. Arms were placed along the body and the legs were stretched. The nose pointed to the ceiling. In preliminary experiments, 5 volunteers tested the device for the possible perception of the wind but could not perceive any related tactile or auditory cue during the swinging movements. Acceleration thresholds were determined by a descending method: The bed was pulled 8 cm backwards and released silently. The subject was asked to notify the operator immediately

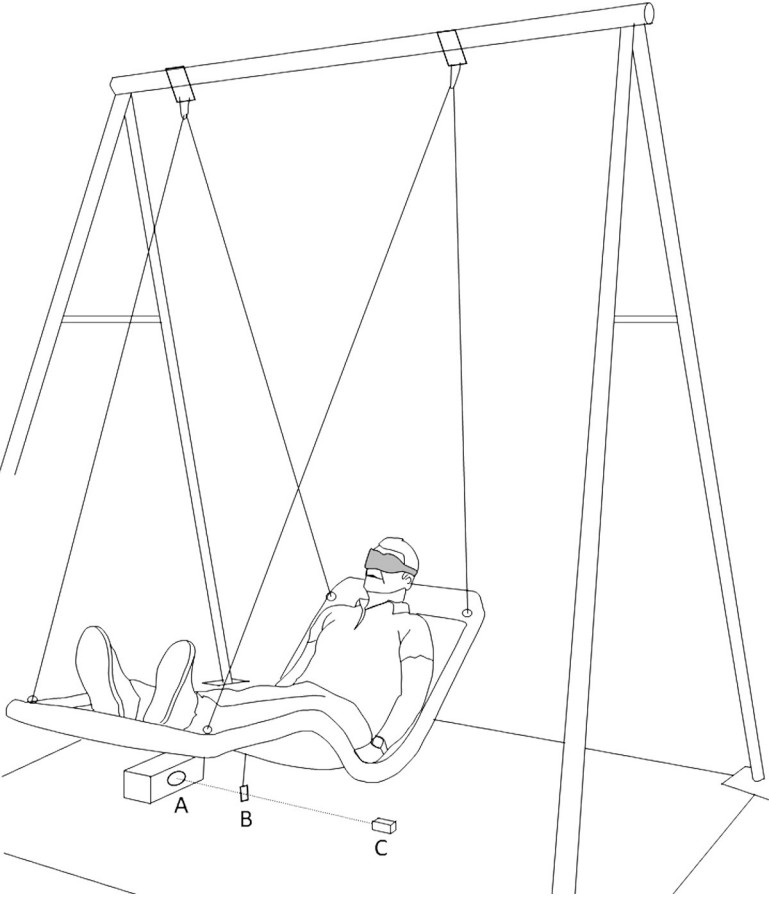

**Fig 1. Experimental setup.** The patient is installed on a swinging bed with eyes blinded by a mask. An electronic device (A) detects the bed's movements and emits a beep at the beginning of each oscillation period. The movement detector is composed of an infrared emitter (A), a deflector (B) and a cover (C) placed under the bed. The cover edge was also used to measure deviation from the equilibrium position in acceleration threshold measurements. The speaker emitting the beep was placed inside the emitter box approximately 1.5 meters away from the subject's ears.

when he/she felt that the bed was immobile. At that time, the operator measured the maximal deviation of the bed from the equilibrium point in cm using the laser projection on the scale placed on the ground. This deviation (d, in meter) was converted to maximal tangential acceleration (a, cm/s$^2$) by the following formula: a = 9.81 X (d/2.4) X 100.

To measure the movement perception delay, we evaluated the range of sound-peak delays which produced a synchronous perception. The bed was pooled backward from its equilibrium point and released. The delays between the beep and the peak were swept from -750 to +750 ms in 50 ms increments. Only backward peaks were used generating only one beep per cycle and allowing a larger time lag exploration. For each lag increment, 3 or more oscillation periods were presented as required by the subject. We chose the peak because it corresponds to the maximum absolute value of deceleration. The peak also corresponds to a change of direction. Describing it to the patients as the "peak" appeared to be easy to understand for the subjects. The range was defined based on preliminary tests to cover the range of synchronous perception delays. The patient was asked to indicate whether the sound and the peak were synchronous. A synchronous perception was noted for a range of delay values defining a synchronous perception interval (SPI, Fig 2). We defined a sound-peak (SP) threshold at the lower limit and a peak-sound (PS) threshold at the upper limit of this interval. Each of these

thresholds was defined by an increment yielding a positive response followed by 2 negative responses to the following increments. For sound-movement synchronicity, each bed release was followed by 8–10 supra liminary oscillations. Each bed release generally allowed testing 2 time-lags. The tolerance of the procedure was evaluated by an auto questionnaire (stress, nausea, discomfort). The mean test duration was approximately 20 minutes.

### Statistical tests

Values were expressed as mean ± Standard Deviation (SD). The normality of the distribution was tested by a Kolmogorov-Smirnov test. Data was analyzed by Graphpad prism (Graphpad Software Inc. V 5.01, La Jolla, CA), Excel (Office, v. 360, Microsoft, Redmond WA) and Statistical Software for social science (SPSS v23, IBM, USA). Test-retest reliability was evaluated by Pearson correlation coefficient R and intraclass correlation coefficient (ICC) assuming two-way random effects, absolute agreement, and a single rater [19]. Internal consistency was evaluated by Cronbach's alpha.

## Results

All subjects perceived the peak of the oscillation (highest position of the head) and the beep as synchronous for a range of delays. The mean width of SPI was 590 ± 193.3 ms (n = 30, S1 Table, Fig 3). The point of subjective synchronicity (PSS, Fig 2) defined as the center of this interval was -244 ± 90.2 ms (n = 30). The sound-peak threshold defined as the upper limit of the SPI was evaluated as 50 ± 90.2 ms (n = 30) and the peak-sound threshold defined as the lower limit of the SPI was estimated as -539±163.8 ms (n = 30). The lower limit of SPI, its middle point (PSS) and width had a relatively small dispersion as evaluated by the relative standard deviation (RSD, 30%, 37%, and 33% respectively). However, the upper limit showed significant variation (RSD = 178%).

The mean acceleration threshold was 9.2 ± 4.60 cm/s$^2$ (n = 30). The dispersion of this measure appeared to be higher than for SPI or PSS (RSD = 87%). There was no correlation between PSS and the acceleration threshold (linear regression test, R = 0.09, p = 0.61).

Parameters concerning the sound-movement delay had a good to excellent reliability in test-retest (same day) and between 2 separate trials (Tables 1 and 2). The acceleration threshold had a lower reliability in the same schedule but still at an acceptable level as judged by Cronbach's alpha and (Table 1, S1 Table), and the Pearson's correlation matrix (Table 2). However, ICC was just below the acceptable level for this parameter (Table 1).

Two subjects complained of nausea after the first trial, and nobody complained of any symptom after the second (including those who complained during the first session). One patient expressed stress for the second trial and anticipated nausea.

## Discussion

In this study, we showed that an oscillatory movement on a swinging bed coupled to a sound signal allows estimating the movement perception delay and acceleration threshold in a non-invasive and reproducible manner in healthy individuals. The estimation of acceleration threshold was in accordance with other reports [1]. The estimation of movement perception delay with a predictable oscillatory movement and a periodic sound stimuli with variable delays led to a synchronous perception of the sound with the peak (maximal head hight) in a relatively wide range of delays (535 ± 190 ms). The negative PSS suggested that generally the sound emitted before the peak was considered as synchronous in this setting.

Many studies have reported on the temporal order of sensory perceptions and the synchronous perception of these inputs [12, 13]. Indeed, the timing of these inputs is of paramount

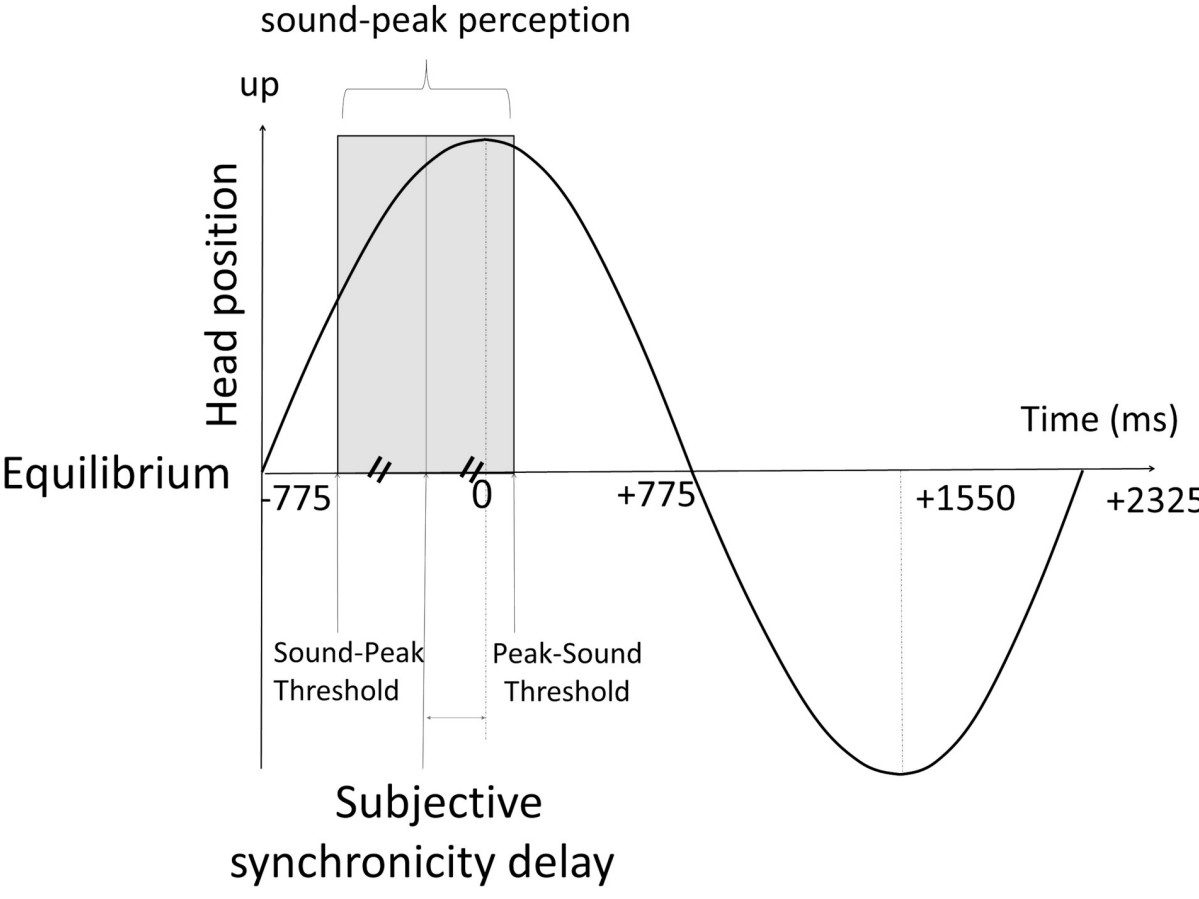

**Fig 2. Relation between sound stimuli and bed oscillation.** During the swinging bed-oscillations a beep was generated by the electronic device with an adjustable time lag. The zero was defined as the peak of the oscillation (head at its maximal height). The time lag was modified from -750 ms to +750 ms with 50 ms increments. Subjects were asked to indicate whether the sound and the peak are synchronous. The synchronous perception interval is depicted in gray. The upper and lower borders were measured. The middle of the range was defined as subjective synchronicity delay.

importance in their coherence during action [13]. Previous works have investigated the delays between vestibular, visual, auditory and sensitive entries and have shown that vestibular sensations are perceived later than sound, vision and touch stimuli [14]. In these works, vestibular galvanic stimulation (GVS) or active head movements were employed to provide a precise time point for the stimuli [13, 14]. In these protocoles, GVS had to occur approximately 160 ms before other stimuli to be perceived as simultanous to them and simple reaction times for perceived head movements were significantly longer to touch, light and sound.

Based on these results, we could have expected a positive PSS (an oscillation peak before sound to be perceived as synchronous). However, the major difference between our protocole and the previous reported results is the predictibility of the swinging movement by the patient and its constant periodicity. In these conditions, we can hypothesize that the mental preparation for the peak perception and its anticipation plays a major role in reducing the delay of its perception. Capacity to synchronise actions and predict timing is essential for movement stability, interaction with environment and respond to unexpected events [20]. Combining multisensory temporal information (vision, hearing, haptic, tactile, and vestibular) for movement

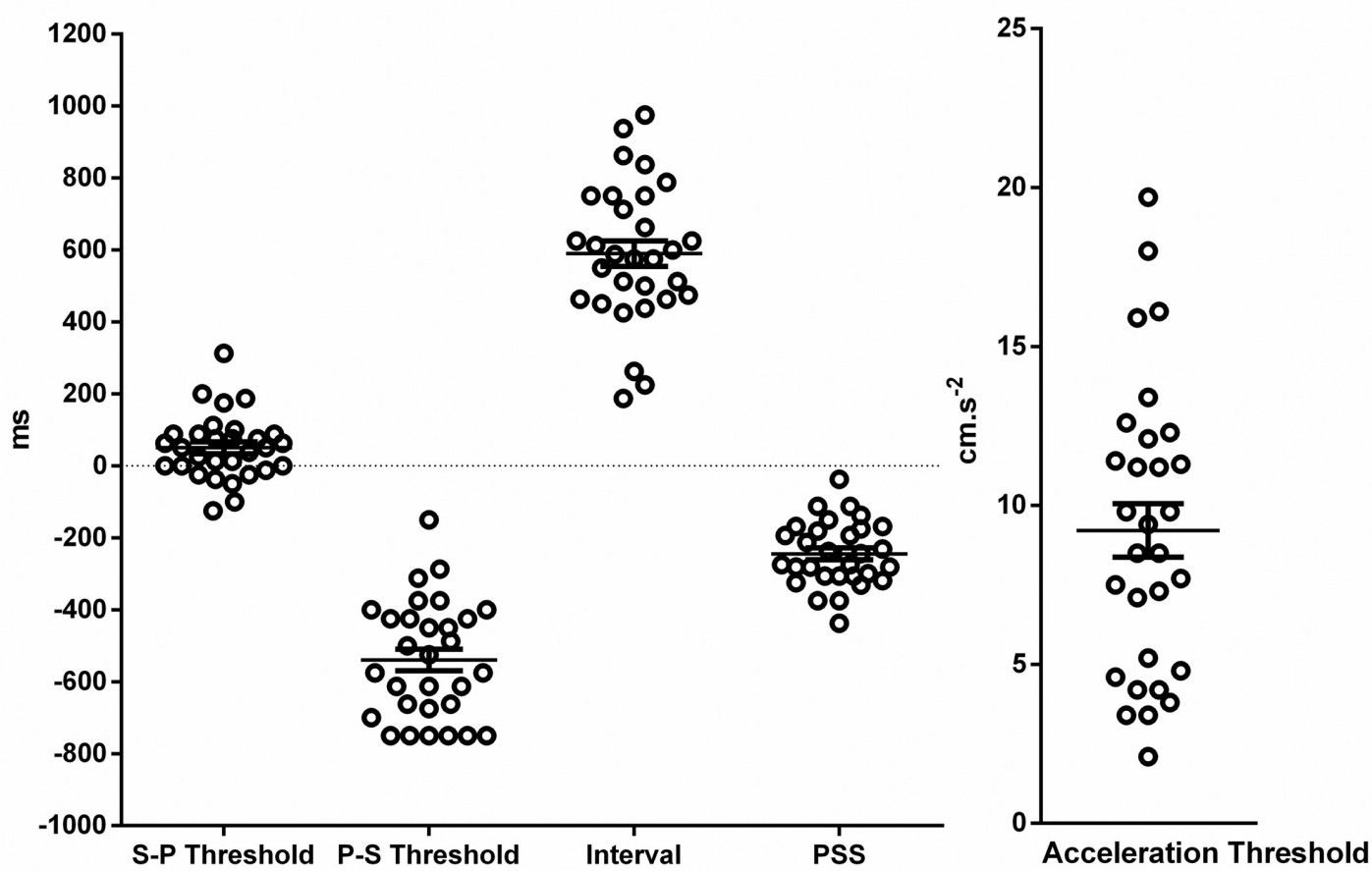

**Fig 3. Dispersion of sound-peak (SP) and peak-sound (PS) thresholds, synchronous perception intervals, point of subjective synchronicity (PSS) and acceleration thresholds.** Open circles represent individual values (n = 30). Each value is the mean of 2 or 4 replicates. Horizontal bars represent mean and the error bars depict standard deviation. For the delays, the zero was defined by the peak of the bed oscillation (head at its maximal height). The point of subjective synchronicity which represents the middle of the synchronous perception interval had a negative value in all cases.

synchronisation and timing has been largely investigated [20]. This synchornization involves prediction and anticipation in rythmic movements such as in music. The movement synchronisation can fit in a linear phase correction model [21] to estimate the temporal corrections

**Table 1. Tau-equivalent reliability (Cronbach's alpha) and intraclass correlation coefficient (ICC) for parameters measured on the swinging bed.**

| Parameter | Cronbach's alpha | Average R | ICC |
|---|---|---|---|
| SP Threshold | 0.92 | 0.74 | 0.76 |
| PS Threshold | 0.81 | 0.54 | 0.39 |
| SSI | 0.89 | 0.68 | 0.68 |
| PPS | 0.89 | 0.67 | 0.62 |
| Acceleration Threshold | 0.75 | 0.42 | 0.46 |

Each parameter was measured in 2 test-retest trials (4 measures for each subject, n = 26). A Cronbach's alpha $\geq 0.7$ was considered as acceptable, $\geq 0.8$ was considered as a good, and $\geq 0.9$ as an excellent internal consistency for the test. R: Correlation coefficient. ICC $<0.50$ was indicative of poor reliability, and an ICC between 0.5–0.75 indicated moderate reliability. SP: sound-peak, PS: peak-sound, PSS: point of subjective synchronicity, SSI: subjective synchronicity interval.

**Table 2. Pearson correlation matrix for test-retest in the 2 trials.**

|  | Retest 1 | Test 2 | Retest 2 |
|---|---|---|---|
| **PSS** |  |  |  |
| **Test 1** | 0.64 [0.36–0.81] *** | 0.51 [0.16–0.75] ** | 0.41 [0.02–0.68] * |
| **Retest 1** |  | 0.66 [0.37–0.84] *** | 0.70 [0.43–0.85] **** |
| **Test 2** |  |  | 0.85 [0.68–0.92] **** |
| **SSI** |  |  |  |
| **Test 1** | 0.80 [0.63–0.90] **** | 0.56 [0.22–0.78] ** | 0.51 [0.15–0.75] ** |
| **Retest 1** |  | 0.69 [0.42–0.85] **** | 0.71 [0.44–0.86] **** |
| **Test 2** |  |  | 0.81 [0.61–0.91] **** |
| **PS threshold** |  |  |  |
| **Test 1** | 0.61 [0.32–0.79] *** | 0.12 [-0.28–0.48] | 0.13 [-0.27–0.49] |
| **Retest 1** |  | 0.38 [0–0.67] | 0.43 [0.05–0.70] * |
| **Test 2** |  |  | 0.78 [0.56–0.90] **** |
| **SP threshold** |  |  |  |
| **Test 1** | 0.78 [0.59–0.88] **** | 0.75 [0.51–0.88] **** | 0.61 [0.30–0.81] **** |
| **Retest 1** |  | 0.79 [0.59–0.90] **** | 0.80 [0.60–0.91] **** |
| **Test 2** |  |  | 0.84 [0.66–0.92] **** |
| **Acceleration threshold** |  |  |  |
| **Test 1** | 0.46 [0.10–0.71] * | 0.67 [0.35–0.85] ** | 0.41 [0.01–0.70] * |
| **Retest 1** |  | 0.51 [0.12–0.77] * | 0.12 [-0.30–0.50] |
| **Test 2** |  |  | 0.50 [0.12–0.76] * |

Values represent Pearson correlation coefficient R [confidence interval] and level of significance:

\* p<0.05

\*\* p<0.01

\*\*\* p<0.001, and

\*\*\*\* p<0.0001. Tests and retests 1 and 2 indicate results from the first and the second trials 2 to 15 days apart. PSS: Point of subjective synchronicity, SSI: subjective synchronicity interval, PS: peak-sound, SP: sound-peak.

made on each movement based on previous asynchrony. Sound appears to have a prominent role among the sensory inputs for movement synchronisation [20].

In our model, subjects faced a multisensorial synchronisation task without a motor reponse. This task requires vestibular and auditory entries as well as a central time-keeping capacity. Similarly to other synchronization tasks with motor response, we hypothesize that subjects estimate and integrate the temporal correction in their multisensory perception and this correction modifies their perception of synchronicity.

Unfortunately, our experimental design does not allow verifying such a hypothesis. With a continuous back-and-forth sweeping of the time lag around the PSS, we could expect a progressive reduction of the SPI or a PSS approaching zero with the increasing number of sweeps.

Another hypothesis to explain our negative PSS (oscillation peak after the sound perceived as synchronous) is that before reaching the peak, the negative acceleration increases rapidly in its absolute value and this phenomenon may contribute to the inverted temporal relation between sound and movement. It would be interesting to study the effetct of the sound emitted at the point of maximum positive acceleration (head at its lowest point) on the PSS.

The reaction delay to movements is crucial for balance. This delay would be the sum of the perception and the response delays. In our study, the protocol was designed in such a way that the subject had ample time (several oscillation periods for each delay) to judge and provide his/her response orally concerning the sound-movement synchronicity. Consequently, the

measured delays do not estimate the reaction time but rather the tolerance of the central integration system for the judgement of synchronicity, and indirectly the movement perception delay.

The precision of the upper and lower limits of the subjective synchronicity interval (SSI), and consequently its center (PSS) depends on the increments. It should be underlined that PSS is not directly signaled by the patient but calculated from the measured upper and lower borders of the SSI which are the sound-peak and the peak-sound thresholds. These thresholds are probably prone to some variations related to the experimental conditions (e.g., bed acceleration, patient's concentration) and this may explain the dispersion of the values. Moreover, while the 50-ms increments allowed us to sweep a large range of delays in a reasonable time, they could limit the precision of the measurements. This could be suspected especially for the upper threshold which has a significant dispersion. Future studies, with smaller time lag increments focusing on the determination of these borders with various paradigms (ascending, descending, and random lags) will be helpful for the standardization of the test.

The vestibular function deteriorates with age [22]. After the age of 60, a reduction in the number of vestibular sensory hair cells, neurons in the scarpa ganglion, and those in the vestibular nuclei is observed [22, 23]. The reduction of otoconia both in number and volume together with alterations of the their compostion are associated to a more frequent detachment of these structures from the otolithic membrane and to changes in the organ function [22–24]. These deteriorations are associated the reduction of vestibulo-ocular reflex gains [22, 23], cervical and ocular vestibular evoked myogenic potentials [25, 26]. This gradual decline potentially particpates in a poorer detection of body movements.

Moreover, the multisensorial integration appears to deteriorate in senior subjects [10, 20]. While the ability of detecting errors in a rythmic sound sequence remains intact with age [27], the multisensory synchronisation of sound and touch [28] or sound and vision [29] become less performant in senior. The effect of aging on sensory synchronisation is not exclusively due to central processing, since the alteration of one of the inputs may also disturb the perceptual synchronisation [20]. In our protocole, it is impossible to distinguish a peripheral deficit from a processing abnormality, but in combination with more peripheral tests, the swinging bed evaluation will potentially provide interesting indications on the mechanisms of dizziness and the risk of falls.

Perceptual acceleration threshold for linear displacements has been already reported in several publications [1, 30–32]. Although authors stated that the movement detection was mainly insured by the otolithic function, experimental setups could not totally supress the tactile and somatosensory cues. The reported acceleration thresholds appeared to be influenced by the stimulation repetition frequency ranging from 1.8 to 8.5 cm/s$^2$ in healthy subjects which is in accordance with our findings. Acceleration threshold measurements had an acceptable test-retest reproducibility, making this parameter a candidate for routine clinical investigation [1]. Interestingly, age also appears to be positively correlated to the anteroposterior acceleration threshold [1], and this observation is in line with the decreased ability of detecting movements in elderly [11].

Our procedure had several limitations. By delivering a pendular movement in a supine position, we stimulated several vestibular captors. The participation of superior and posterior semicircular canals, and both utricular and saccular maculae in the detection of the pendular movement is probable since the acceleration has horizonal, vertical and rotatory components in the vertical plane of the oscillation. Moreover, even if all sensory inputs in exception of auditory and vestibular entries were minimized, the presence of other cues such as somatosensory information could not be excluded at supraliminary stimulation levels. The presence of uncontrolled sensory cues would increase the intraindividual and inter individual variabilities of the

parameters, but our measures appeared reproducible and coherent indicating the stability of the sensory cues during the trials. Further studies will probably elucidate the participation of different inputs and central processing in this test. Another issue is the detection of the sound source movement relative to the head position by monaural (spectral changes, doppler effect) [33] or binaural functions (interaural time and intensity differences) [34]. These effects are sensitive to signal duration [35]. Considering the shortness of the sound stimuli (5 ms, 0.16% of the oscillation period), this effect can be considered as negligeable.

The advantages of this system in comparison to what is presented in the literature to estimate the movement perception is that the swinging bed remains easy to install and calibrate and appears non-invasive. It is applicable to fragile, handicaped, senior subjects and children. Instructions are easy to understand and the test procedure is relatively short.

## Conclusion

Our swinging bed coupled to a sound source provided reproducible and coherent perceptual acceleration threshold and movement perception delay in healthy human subjects. This non-invasive and simple device potentially allows exploring otoneurological diseases and fallers.

## Supporting information

**S1 Table. Individual data concerning swinging bed measurements and test tolerance.** Acceleration thresholds are expressed as swinging bed deviations in cm. This deviation was converted to maximal tangential acceleration (a, cm/s$^2$) by the following formula: a = 9.81 X (d/240) X 100.
(XLSX)

**S1 Appendix. Individual data concerning swinging bed measurements and test tolerance.** Acceleration thresholds are expressed as swinging bed deviations in cm. This deviation was converted to maximal tangential acceleration (a, cm/s$^2$) by the following formula: a = 9.81 X (d/240) X 100.
(XLSX)

## Author Contributions

**Conceptualization:** Michel Toupet, Alexis Bozorg Grayeli.

**Data curation:** Maxime Guyon, Cyrielle Chea, Davy Laroche.

**Formal analysis:** Maxime Guyon, Davy Laroche, Alexis Bozorg Grayeli.

**Investigation:** Cyrielle Chea, Audrey Baudet, Michel Toupet, Alexis Bozorg Grayeli.

**Methodology:** Davy Laroche, Isabelle Fournel, Audrey Baudet, Michel Toupet, Alexis Bozorg Grayeli.

**Project administration:** Alexis Bozorg Grayeli.

**Software:** Audrey Baudet.

**Supervision:** Isabelle Fournel.

**Validation:** Alexis Bozorg Grayeli.

**Writing – original draft:** Maxime Guyon, Davy Laroche, Michel Toupet, Alexis Bozorg Grayeli.

**Writing – review & editing:** Alexis Bozorg Grayeli.

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
