## [Decision Letter · Decision Letter 0]

25 Mar 2021

PONE-D-20-17020

Measuring Threshold and Latency of Motion Perception on a Swinging Bed

PLOS ONE

Dear Dr. Bozorg Grayeli,

Thank you for submitting your manuscript to PLOS ONE. After careful consideration, we feel that it has merit but does not fully meet PLOS ONE’s publication criteria as it currently stands. Therefore, we invite you to submit a revised version of the manuscript that addresses the points raised during the review process.

Your paper presents and evaluates a new method for assessing motion perception. This test is interesting because it is quick, simple and not expensive.

However, reviewers highlighted some important shortcomings that need to be addressed. The main one is that reproduciblilty has been evaluated at the group level not at indivual level but you need to pay carefull attention to the other concerns. Thus this paper need major revision,

We look forward to receiving your revised manuscript.

Kind regards,

Pierre Denise, Ph.D, M.D.

Academic Editor

PLOS ONE

Journal Requirements:

2. We note that some of the images may contain depictions of humans. Authors submitting manuscripts that include identifying or potentially identifying information must comply with our requirements for informed consent (https://journals.plos.org/plosone/s/human-subjects-research#loc-patient-privacy-and-informed-consent-for-publication).

3. Thank you for including your ethics statement:  "The protocol was reviewed and approved by the institution’s ethical committee (CPP Est III, France) and a written consent was obtained from all subjects. We have complied with APA ethical standards in the treatment of the subjects.".   

5. We note that Figure 1 includes an image of a participant in the study. 

Reviewers' comments:

Reviewer's Responses to Questions

**Comments to the Author**

1. Is the manuscript technically sound, and do the data support the conclusions?

Reviewer #1: Yes

Reviewer #2: No

2. Has the statistical analysis been performed appropriately and rigorously? 

Reviewer #1: Yes

Reviewer #2: No

3. Have the authors made all data underlying the findings in their manuscript fully available?

Reviewer #1: Yes

Reviewer #2: No

4. Is the manuscript presented in an intelligible fashion and written in standard English?

Reviewer #1: Yes

Reviewer #2: Yes

5. Review Comments to the Author

Reviewer #1: Reviewing article PoNe-D-2017020

Measuring Threshold and latency of Motion Perception on a Swinging Bed

I have read with interest this article which deals with a prospective study performed in 30 healthy blindfolded subjects explored with a rehabilitation swinging bed to evaluate and measure latency and threshold of pendular motion perception. This study was conducted in a population aged 20-61.

The mean acceleration threshold is given at 9.2±4.6 °/s – 2. For the latency estimation the point of subjective synchronicity defined as the center of the range width of the synchronous perception interval is given at -195±106 ms

In this work authors show after test and retest that this procedure is reliable to measure in a non-invasive manner information on movement perception. They suggest possible application for exploration in aging patients.

The originality of this work lies in studying a double task with concomitant sound and vestibular information making this method a tool particularly devoted to the study of aging people

This work is well presented and shows a promising procedure to explore with a non invasive, simple and not expensive method older subjects in order to identify alterations of motion perception threshold and modification of latency response related with ageing

Minor remarks

Abstract : possible typing error in the “”results” paragraph given by PLUs one “9.2±4.60 cm.s-2” which is not found and seems corrected in the final abstract and correctly written line 34 “. A similar typing error is noted p 12 line 227 to indicate -2. The indication for accelerations should be standardized for example cm/s-2 or else but written everywhere similarly

Introduction: line 61 typing error “…inputs are processed more rapidly that vestibular..”: “that” should be replaced by “than”

Fig 1 line 104: “The patient..with closed eyes..” “”closed eyes” should be replaced by “eyes blinded by a mask” which seems to be more accurate

Discussion: line 194”…GVS had to occure..” should be “had to occur”

Discussion line 215: “While the ability..sequence remais intact…”: should be “… remains..”

It could be suggested for more clarity and to easy the reading to give at the beginning of this work a short list of abbreviations

PSS; GVS; SPI: sound perceptual interval; SSI; SSD..And so on..

Remarks

The age of the studied population ranges from 20 to 61. The measure of the latency is established on the response after a sound to indicate the synchronicity or not of the beep/peak of each oscillation. The beep is dispatched by a loud speaker and the distance of the sound source to the examined subject should be specified . The sound velocity in the air is around 300 m/s , so if the source is at 1 or 2 or 3 m the delay of the sound to be perceived is around 3 to 10 ms The question of a beep displayed by a ear phone could possibly rule out a possible Bias related to the protocol and not directly related to the subject response. What is the rationale to have preferred a loud speaker to earphones?

The authors have opted for a vocal signal over a manual signal for indicating the concordance resented by the patient with their perception and the beep emission . In this way the reaction time delay is reasonably minimized

It would have been interesting to have in the discussion considerations about the reaction time which is higher in aging peoples over 60 or 65 Yo.

On one other hand Otolith system performances explored with VEMP shows modifications with aging and the discussion could have consider with more details and more clarity the possible contribution of the different vestibular systems to a possible modification of the threshold of motion perception and latency . However authors mention in the chapter limitations that in their procedure the supine position stimulate several vestibular captors . these different captors could be possibly cited including otolith, proprioceptive but also probably superior semicircular canals since in this procedure only the contribution of the lateral SCC seems to be ruled out.

Other Remarks

The introduction should be slightly modified since it is advocated and presented as a work about aging…So it is expected to find results about measuring perception delay in aging patient...; However the results are only given for patients from 20 to 61 y o ? . It would have been more logical to present clearly and simply this work as a preliminary work for studying in a further work aging peoples and to present the main goal of this study to verify the validity of this model of experiment to evaluate with a great reliability and simplicity with a not expensive material latency of motion perception and the function of the otolith or a more global vestibular or balance system ?

The decline of the otolithic system is signalled in aging people by Y. Agrawal , F. Wuyts et al (JV res 2019) and by Lingchao Li (2018) when measuring the cVEMP as soon as 60 YO and for oVEMP a little later (60 -80 YO ) by Li et al (2015) and Tseng et al (2010) ; degenerescence of otoconies is mentioned by Rosenhall et al and JI S and Zhai (2018). Baltes et al and Baloh et al have signaled that 50% of adults older than 60 YO have a physiological impairment of their physiological function. These aspects could be more developed in the discussion .

One interesting point is the dispersion of sound-peak and peak –sound thresholds and synchronous perception intervals: the point of subjective synchronicity is signaled before the real point of the peak (head at the maximal height) possibly correlated to sensitivity of the threshold and could be more largely commented in the discussion.

This work is interesting and provides a suitable and promising method to evaluate in a simple way modifications of motion perception threshold and latency for further studies evaluating aging peoples. It could be accepted for publication after minor modifications concerning the introduction, explanations of a few options in the protocol (source of emission of beeps) and in the discussion a more clear exposition of the possible inner ear or proprioceptive targets involved to explain the possible modifications expected in aging peoples in a further study

Reviewer #2: In this paper, the authors tested the possibility to use a new, simple set-up for both estimating the acceleration threshold for perceiving body oscillations and for assessing the latency of motion perception. A secondary objective was to determine whether the results obtained with this set-up were reproducible. The set-up consisted of a rehabilitation swinging bed suspended to a 2.5 m-high gantry. Using this set-up, the authors found an acceleration threshold similar to what is reported in the literature. However, the latency of motion perception appeared different to what was expected.

There is indeed a need for developing a simple, and relative low-cost set-up for exploring otoneurological diseases. This effort is therefore a welcome one. Unfortunately, the description of the set-up, and of the methods are not sufficiently detailed to judge the validity of the study and of the device. The methods section is not written with enough information so that the experiment could be repeated by others.

From my understanding of the methods section (note that I might be wrong), only two trials were performed to test the reproducibility of the results related to the acceleration threshold for perceiving body oscillations. The authors tested the reproducibility at the group level, rather than at the individual level. Because the authors’ goal was to develop and to evaluate a new system, a thorough test of the reproducibility is needed. Rather, the authors tried to keep the experimental test as short as possible, as if they were doing clinical testing.

Specific comments

Line 28: change “patients” by “subjects” (throughout the text).

Line 53 : It should be made clear that the 2 parameters presented here (based on body acceleration) are not the only parameters that can characterize the awareness of the body movement.

Line 63: The authors should provide some examples of methods that are currently used to test the awareness of the body movements by vestibular inputs and specify why they judged them unsatisfactory.

Line 66: A minimum of information should be provided about rehabilitation swinging beds to understand why they can be considered as potentially efficient for the exploration of movement perception. It is not clear why the authors feel swinging beds safer and less invasive than methods currently used for the exploration of movement perception. For instance, Kingma (2005), cited by the authors, used “a motor driven linear sled running on a horizontal track of 4.2 metres (maximum velocity 3.7 m/s; maximum acceleration 1.2 m/s2 adjustable in steps of 1 cm/s2” (page 2). “The subjects were seated upright with their feet on a footrest; head fixed against a headrest and the body restrained with safety belt” (page 3). This method then appears perfectly safe and non-invasive.

Line 67. The first hypothesis presented at the end of the introduction “the comparison between the delay of a sound stimulus and the body oscillation on a swinging bed could be reproducible parameter to estimate the multisensory integration of movement perception” is not clear and needs to be rephrased. Moreover, the fact that this hypothesis is related to multisensory integration came a bit as a surprise as the authors did not discuss about multisensory integration in the introduction.

Line 69: The sentence presenting the second hypothesis is poorly formulated: “We also hypothesized that the acceleration perception threshold could be measured on the same device”. I also wonder if this can be really considered as an experimental hypothesis.

Line 72: The authors mentioned that their aim “was to develop a system to measure the delay of body movement perception and the threshold of acceleration perception”. At this point, the difference between “body movement perception” and “threshold of acceleration perception” is not clear.

Line 94: What are the sensory systems stimulated by the bed oscillations?

Line 98: What was the spatial resolution of the scale?

Line 98: The methods used to measure bed motion is difficult to understand. It seems that the device could detect bed position/movement as it produced, for each cycle, a beep when the subject’s head was at its highest position. Why then was it necessary to use the projection of the laser on the scale on the ground to estimate head tangential acceleration? How this device allowed sending a beep precisely 750 ms (and 700ms, 650 ms, etc.) before the head reached its highest position?

Line 99: Even after several readings, it is hard to understand the methods used to measure the latency of the movement perception. The text indicates that an infrared detector was placed on the ground to detect the passage of the bed at its lowest point at each cycle. This device was connected to a processor and a loudspeaker enabling the system to produce a beep when the patient’s head was at its highest position. There is something missing to understand how the signal detected at ground level can be used to produce a beep when the patient head was at its highest position.

Line 112: Which method was used in preliminary experiments to test a possible effect of wind during the swing movement?

Line 116: From my understanding of the methods, for each subject, acceleration thresholds were only tested twice. If this was indeed the case, it is not enough to assess the reproducibility of the results, particularly because the aim of this study was to assess this reproducibility.

Line 116: “all measurements”. Are there many? The authors should specify what these measurements are.

Line 119: The methods used to measure the movement perception delay need clarification. Were both forward and backward peaks used for this assessment? If so, did movement perception differ according to the considered peak? How many oscillations were produced by each bed release? How many bed releases were needed for assessing movement perception? The oscillation amplitudes decreased with the number of cycles. Were the oscillations stopped when their amplitude dropped below a given amplitude, before producing a new bed release? Also, why were the subjects asked to estimate the time their head reached their highest position rather than to estimate the time of the downward motion onset? The latter variable would seem more appropriate for estimating movement perception.

Line 152: Does the authors have an explanation for the observed large difference between the sound-peak threshold and the peak-sound threshold?

Line 185: The sound had to occur before the peak to be considered as synchronous. On the contrary, and as mentioned by the authors, previous studies showed that vestibular sensations were perceived later than sounds. The authors proposed that this could be due to the predictability of the swinging movement in their study, while vestibular stimulation could not be predicted in previous studies. This hypothesis is plausible. Another hypothesis that needs to be considered is that prior to the peak, body acceleration fell below the acceleration threshold for detecting body motion. The authors’ finding could also result from a delay for generating the beep on the basis of bed position signal. It is important to measure and to provide this delay (and as specified above, to specify the methods used to precisely send to beeps prior to the peaks).

Line 210: The authors hypothesized that subjects made temporal corrections based on previous asynchrony. This could be tested by testing whether the asynchrony changed over the experimental session.

Line 234: The authors mentioned that the swinging bed stimulated non-vestibular sensory inputs (e.g. somatosensory inputs) before adding “Nevertheless, the measures were reproducible and appeared to be coherent”. What is the link between these two statements?

Figure 3 caption. Sound-peak (SP) and peak-sound (PS) thresholds should be defined in the text.

Figure 3. The figure shows that 6 subjects had the same P-S threshold, probably -750 ms, i.e. the greatest lag used in the study. I wonder if these data represent the actual subjects’ perception of the synchrony between the sound and the peak.

6. PLOS authors have the option to publish the peer review history of their article (what does this mean?). If published, this will include your full peer review and any attached files.

Reviewer #1: No

Reviewer #2: No

---

## [Author Response · Author response to Decision Letter 0]

17 May 2021

Response to reviewers

Reviewer 1

I have read with interest this article which deals with a prospective study performed in 30 healthy blindfolded subjects explored with a rehabilitation swinging bed to evaluate and measure latency and threshold of pendular motion perception. This study was conducted in a population aged 20-61.

The mean acceleration threshold is given at 9.2±4.6 °/s – 2. For the latency estimation the point of subjective synchronicity defined as the center of the range width of the synchronous perception interval is given at -195±106 ms.

In this work, authors show after test and retest that this procedure is reliable to measure in a non-invasive manner information on movement perception. They suggest possible application for exploration in aging patients.

The originality of this work lies in studying a double task with concomitant sound and vestibular information making this method a tool particularly devoted to the study of aging people

This work is well presented and shows a promising procedure to explore with a non-invasive, simple and not expensive method older subjects in order to identify alterations of motion perception threshold and modification of latency response related with ageing

Minor remarks

Q1- Abstract : possible typing error in the “”results” paragraph given by PLUs one “9.2±4.60 cm.s-2” which is not found and seems corrected in the final abstract and correctly written line 34 “. A similar typing error is noted p 12 line 227 to indicate -2. The indication for accelerations should be standardized for example cm/s-2 or else but written everywhere similarly.

R1- We have now homogenized and cm/s2 was adopted throughout the text.

Q2- Introduction: line 61 typing error “…inputs are processed more rapidly that vestibular..”: “that” should be replaced by “than”.

R2- This error is now corrected.

Q3- Fig 1 line 104: “The patient..with closed eyes..” “”closed eyes” should be replaced by “eyes blinded by a mask” which seems to be more accurate.

R3- This sentence is now rectified.

Q4- Discussion: line 194”…GVS had to occure..” should be “had to occur”

R4- This spelling error is now corrected.

Q5- Discussion line 215: “While the ability..sequence remais intact…”: should be “… remains..”

R5- This typo is now corrected.

Q6- It could be suggested for more clarity and to easy the reading to give at the beginning of this work a short list of abbreviations PSS; GVS; SPI: sound perceptual interval; SSI; SSD..and so on..

R6- A list of abbreviations is now provided at the beginning of the revised manuscript. Several misused abbreviations are also corrected in the text.

Remarks

Q7- The age of the studied population ranges from 20 to 61. The measure of the latency is established on the response after a sound to indicate the synchronicity or not of the beep/peak of each oscillation. The beep is dispatched by a loud-speaker and the distance of the sound source to the examined subject should be specified. The sound velocity in the air is around 300 m/s, so if the source is at 1 or 2 or 3 m the delay of the sound to be perceived is around 3 to 10 ms, the question of a beep displayed by an earphone could possibly rule out a possible Bias related to the protocol and not directly related to the subject response. What is the rationale to have preferred a loudspeaker to earphones?

R7- The rational was to prevent any perception of movement related to the wires connected to those earphones. Moreover, earphones would have hampered the dialogue between the subject and the examiner. We did not choose wireless headphones (Bluetooth) since they show a 32 ms lag in optimal conditions. The loudspeaker was placed inside the infrared detection box under the swinging bed and approximately 1.5 m from the patient’s ears. This point is now added in “the materials and methods” section and in Fig. 1.

Q8- The authors have opted for a vocal signal over a manual signal for indicating the concordance presented by the patient with their perception and the beep emission. In this way the reaction time delay is reasonably minimized. It would have been interesting to have in the discussion considerations about the reaction time which is higher in aging peoples over 60 or 65 Yo.

R8- This is indeed a very interesting point. The reaction delay to movements is crucial for balance. This delay would be the sum of the perception and the response delays. In our study, the protocol was designed in such a way that the subject had ample time to judge and provide his/her response orally concerning the sound-movement synchronicity. This point is now clarified in the materials and methods section as follows “For each increment, 3 or more oscillation periods were presented as required by the subject.” Consequently, the measured delays do not estimate the reaction time but rather the tolerance of the central integration system for the judgement of synchronicity, and indirectly the movement perception delay. This point is now added to the discussion (lines 278-284).

Q9- On one other hand Otolith system performances explored with VEMP shows modifications with aging and the discussion could have consider with more details and more clarity the possible contribution of the different vestibular systems to a possible modification of the threshold of motion perception and latency. However, authors mention in the chapter limitations that in their procedure the supine position stimulate several vestibular captors. These different captors could be possibly cited including otolith, proprioceptive but also probably superior semicircular canals since in this procedure only the contribution of the lateral SCC seems to be ruled out.

R9- This important point is now added to the discussion, and the captors are cited as follows (lines 323-327): “By delivering a pendular movement in a supine position, we stimulated several vestibular captors. The participation of superior and posterior semicircular canals, and both utricular and saccular maculae in the detection of the pendular movement is probable since the acceleration has horizonal, vertical and rotatory components in the vertical plane of the oscillation.” 

Other Remarks

Q10- The introduction should be slightly modified since it is advocated and presented as a work about aging…So it is expected to find results about measuring perception delay in aging patient...; However the results are only given for patients from 20 to 61 y o? . It would have been more logical to present clearly and simply this work as a preliminary work for studying in a further work aging peoples and to present the main goal of this study to verify the validity of this model of experiment to evaluate with a great reliability and simplicity with a not expensive material latency of motion perception and the function of the otolith or a more global vestibular or balance system?

R10- The introduction is now modified to set the background for the present study in the first paragraph. The application of the system in aging population is now presented as a perspective in the second paragraph.

Q11- The decline of the otolithic system is signaled in aging people by Y. Agrawal, F. Wuyts et al (JV res 2019) and by Lingchao Li (2018) when measuring the cVEMP as soon as 60 YO and for oVEMP a little later (60 -80 YO) by Li et al (2015) and Tseng et al (2010); degenerescence of otoconies is mentioned by Rosenhall et al and JI S and Zhai (2018). Baltes et al and Baloh et al have signaled that 50% of adults older than 60 YO have a physiological impairment of their physiological function. These aspects could be more developed in the discussion.

R11- A paragraph describing the decline of the vestibular system is now added to the discussion. Thank you for providing these interesting references. We have now included most of them in this section as follows (lines 296-303): “The vestibular function deteriorates with age [22]. After the age of 60, a reduction in the number of vestibular sensory hair cells, neurons in the scarpa ganglion, and those in the vestibular nuclei is observed [22, 23]. The reduction of otoconia both in number and volume together with alterations of the their compostion are associated to a more frequent detachment of these structures from the otolithic membrane and to changes in the organ function [22-24]. These deteriorations are associated the reduction of vestibulo-ocular reflex gains [22, 23], cervical and ocular vestibular evoked myogenic potentials [25, 26]. This gradual decline potentially particpates in a poorer detection of body movements.”

Q12- One interesting point is the dispersion of sound-peak and peak-sound thresholds and synchronous perception intervals: the point of subjective synchronicity is signaled before the real point of the peak (head at the maximal height) possibly correlated to sensitivity of the threshold and could be more largely commented in the discussion.

R12- It is indeed very interesting to study the dispersion of these values and to understand the reason why PSS has a negative value. The precision of the upper and lower limits of the subjective synchronicity interval (SSI), and consequently its center (PSS) depends on the increments. It should be underlined that PSS is not directly signaled by the patient but calculated from the measured upper and lower borders of the SSI which are the sound-peak and the peak-sound thresholds. These thresholds are probably prone to some variations related to the experimental conditions (e.g., bed acceleration, patient’s concentration) and this may explain the dispersion of the values. Moreover, while the 50-ms increments allowed us to sweep a large range of delays in a reasonable time, they could limit the precision of the measurements. This could be suspected especially for the upper threshold which has a significant dispersion. Future studies, with smaller time lag increments focusing on the determination of these borders with various paradigms (ascending, descending, and random lags) will be helpful for the standardization of the test. These points are now added to the discussion (lines 286-295). 

Q13- This work is interesting and provides a suitable and promising method to evaluate in a simple way modifications of motion perception threshold and latency for further studies evaluating aging peoples. It could be accepted for publication after minor modifications concerning the introduction, explanations of a few options in the protocol (source of emission of beeps) and in the discussion a more clear exposition of the possible inner ear or proprioceptive targets involved to explain the possible modifications expected in aging peoples in a further study.

R13- Thank you for your constructive remarks. We have taken them all into account in the revised version.

 

Reviewer 2

In this paper, the authors tested the possibility to use a new, simple set-up for both estimating the acceleration threshold for perceiving body oscillations and for assessing the latency of motion perception. A secondary objective was to determine whether the results obtained with this set-up were reproducible. The set-up consisted of a rehabilitation swinging bed suspended to a 2.5 m-high gantry. Using this set-up, the authors found an acceleration threshold similar to what is reported in the literature. However, the latency of motion perception appeared different to what was expected.

Q- There is indeed a need for developing a simple, and relative low-cost set-up for exploring otoneurological diseases. This effort is therefore a welcome one. Unfortunately, the description of the set-up, and of the methods are not sufficiently detailed to judge the validity of the study and of the device. The methods section is not written with enough information so that the experiment could be repeated by others.

R- Following your suggestions and those of the first reviewer, we have provided all the technical details concerning the material and the procedure.

Q1- From my understanding of the methods section (note that I might be wrong), only two trials were performed to test the reproducibility of the results related to the acceleration threshold for perceiving body oscillations. The authors tested the reproducibility at the group level, rather than at the individual level. Because the authors’ goal was to develop and to evaluate a new system, a thorough test of the reproducibility is needed. Rather, the authors tried to keep the experimental test as short as possible, as if they were doing clinical testing.

R1- In this study, we conducted not 2 but 4 tests for each parameter and each subject. A test-retest with a 10 min. interval, and a second test-retest on the same group 2 to 15 days after the first. These tests were conducted for both the sound movement synchronicity and the acceleration threshold. This information was provided in the materials and methods. To clarify this point we have now added the following sentence at the beginning of the paragraph: “A total of 4 tests was designed for each subject and each parameter.” 

Indeed, the reproducibility was evaluated at the group level by Cronbach’s alpha. We have now added new analyses for test reliability at individual level by providing the intraclass correlation coefficient (revised Table 1) the Pearson correlation matrix (Table 2). 

One trial (test-retest) took approximately one hour including 20 minutes for each test, a 10-minute break and 10 minutes for explanations and questionnaire. The concern of limiting its duration was motivated by the idea that repeating and increasing the duration of such psychophysical tests may alter the reliability by a lack of concentration or tolerance to movements. 

Specific comments

Q2- Line 28: change “patients” by “subjects” (throughout the text).

R2- This modification has been applied throughout the text.

Q3- Line 53: It should be made clear that the 2 parameters presented here (based on body acceleration) are not the only parameters that can characterize the awareness of the body movement.

R3- Indeed, several other factors such visual perceptive and auditory cues, motor responses, movement intention can also participate in the awareness and its characterization. This part has now been developed as follows (lines 79-83): “This awareness can be characterized by several parameters (e.g., change of direction relative to the gravity vector, relative movement of body parts, change of location in space) among which, the perception threshold of body acceleration and the delay of this perception are interesting. Indeed, the impaired perception of fall timing appears to be related to the risk of fall in the elderly [9].”

Q4- Line 63: The authors should provide some examples of methods that are currently used to test the awareness of the body movements by vestibular inputs and specify why they judged them unsatisfactory.

R4- Following your suggestion, we have now added the following examples to the first paragraph of the introduction: “To evaluate the perception of circular movements in healthy subjects, Nooij et al. employed a MPI Cybermotion Simulator [2]. Sensitivity to vertical self-motion was evaluated in healthy volunteers on a similar device by Nesti et al [3]. Other authors set up a Moog motion platform to detect dynamic tilt thresholds in patients with vestibular migraine [4] or a motor-driven linear sled on a 4.2-m track to assess linear movement perception [1]. The complexity of the setups, the duration of the examination, their cost and cumbersomeness hamper their clinical use in routine.”

Q5- Line 66: A minimum of information should be provided about rehabilitation swinging beds to understand why they can be considered as potentially efficient for the exploration of movement perception. It is not clear why the authors feel swinging beds safer and less invasive than methods currently used for the exploration of movement perception. For instance, Kingma (2005), cited by the authors, used “a motor driven linear sled running on a horizontal track of 4.2 metres (maximum velocity 3.7 m/s; maximum acceleration 1.2 m/s2 adjustable in steps of 1 cm/s2” (page 2). “The subjects were seated upright with their feet on a footrest; head fixed against a headrest and the body restrained with safety belt” (page 3). This method then appears perfectly safe and non-invasive.

R5- Thank you for raising this interesting point. By the word “efficient” we intended to underline the fact that we might obtain similar information on movement perception with a less complex setup and a shorter examination time. Although, none of the reported setups appeared dangerous in healthy volunteers, it might be more difficult to apply some of them to fragile or dizzy patients. To our knowledge and in contrast to physiotherapy swinging beds, safety studies have not been carried on these experimental platforms and they do not meet requirements for routine clinical use. We have now clarified this point by suppressing the words “less invasive” and by adding the above explanation in the introduction (lines 59-69).

Q6- Line 67: The first hypothesis presented at the end of the introduction “the comparison between the delay of a sound stimulus and the body oscillation on a swinging bed could be reproducible parameter to estimate the multisensory integration of movement perception” is not clear and needs to be rephrased. Moreover, the fact that this hypothesis is related to multisensory integration came a bit as a surprise as the authors did not discuss about multisensory integration in the introduction.

R6- According to your suggestion, we have now modified the paragraph as follows: “Multisensory integration of visual, vestibular, proprioceptive, and auditory cues for movement perception is crucial in balance and seems to be affected by diseases such as vestibular migraine [Mahoney et al.] or age [Versino et al.]. We hypothesized that this integration could be assessed by exploring the synchronous perception of a sound and a passive body oscillation on a swinging bed.”

Q7- Line 69: The sentence presenting the second hypothesis is poorly formulated: “We also hypothesized that the acceleration perception threshold could be measured on the same device”. I also wonder if this can be really considered as an experimental hypothesis.

R7- We agree that this sentence is too short and based on unexplained assumptions. We have now modified this part as follows: “Measuring acceleration perception threshold has potential implications on understanding the mechanisms of dizziness and fall [Richerson et al., 2020]. Threshold values are subject to significant variation depending on the plane of the stimulation and stimulus profile (sinus, linear, steps, etc.) [Kingma, 2005]. We hypothesized that we could measure a reproducible threshold on the swinging bed during deceleration. From a practical standpoint, measuring 2 potentially important parameters (synchronous perception of sound and movement and acceleration perception threshold) on the same device and with the same setup would be interesting in a clinical setup.”

Q8- Line 72: The authors mentioned that their aim “was to develop a system to measure the delay of body movement perception and the threshold of acceleration perception”. At this point, the difference between “body movement perception” and “threshold of acceleration perception” is not clear.

R8- To clarify the aim, we have now changed this paragraph to: “The aim of this study was to develop a system to measure the delays for which sound and body movement were perceived as synchronous, and the threshold of acceleration perception on a safe device applicable to clinical routine and to evaluate its tolerance and reliability in healthy adults.”

Q9- Line 94: What are the sensory systems stimulated by the bed oscillations?

R9- In this pendular oscillation, probably both semicircular canals and otolithic organs are stimulated. Visual, tactile, and unwanted auditory cues were suppressed or negligeable, but visceroceptive inputs were probably present. This part is now detailed in the discussion (lines 324-329) as follows: “Our procedure had several limitations. By delivering a pendular movement in a supine position, we stimulated several vestibular captors. The participation of superior and posterior semicircular canals, and both utricular and saccular maculae in the detection of the pendualr movement is probable since the acceleration has horizonal, vertical and rotatory components in the vertical plane of the oscillation. Moreover, even if all sensory inputs in exception of auditory and vestibular entries were minimized, the presence of other cues such as somatosensory information could not be excluded at supraliminary stimulation levels.”

Q10- Line 98: What was the spatial resolution of the scale? 

R10- The scale had a millimetric resolution. This point is now added to the Materials and Methods (line 130).

Q11- Line 98: The methods used to measure bed motion is difficult to understand. It seems that the device could detect bed position/movement as it produced, for each cycle, a beep when the subject’s head was at its highest position. Why then was it necessary to use the projection of the laser on the scale on the ground to estimate head tangential acceleration? 

R11- To measure the acceleration threshold, the operator let the bed come to a stop progressively. During this phase, when the subject announced that he/she did not perceive any movement, the bed continued to oscillate slightly. Only at this moment, the laser beam on the scale placed on the ground was used to measure the maximum deviation of the bed from its equilibrium point. This distance from the point of equilibrium in cm was then converted to acceleration in cm/s2. We apologize for this lack of clarity. We have now completed the explanation as follows (lines 151-156): “The bed was pulled 8 cm backwards and released silently. The subject was asked to notify the operator immediately when he/she felt that the bed was immobile. At that time, the operator measured the maximal deviation of the bed from the equilibrium point in cm using the laser projection on the scale placed on the ground. All measurements were repeated twice in a test-retest design. This deviation (d, in meter) was converted to maximal tangential acceleration (a, cm/s2) by the following formula: a=9.81 X (d/2.4) X 100”

Q12- Line 98: How this device allowed sending a beep precisely 750 ms (and 700ms, 650 ms, etc.) before the head reached its highest position?

R12- During the first 3 bed passages in front of the infrared detector (half cycles), the device measured and averaged the half cycles of the oscillation. Then, the system began to emit a beep with a negative or a positive time lag based on this calculated period. This clarification is now added to materials and methods (lines 135-138 of the revised manuscript).

Q13- Line 99: Even after several readings, it is hard to understand the methods used to measure the latency of the movement perception. The text indicates that an infrared detector was placed on the ground to detect the passage of the bed at its lowest point at each cycle. This device was connected to a processor and a loudspeaker enabling the system to produce a beep when the patient’s head was at its highest position. There is something missing to understand how the signal detected at ground level can be used to produce a beep when the patient head was at its highest position.

R13- As explained above, the device measured the oscillation period over 3 half cycles, and then produced an anticipated or a delayed beep rhythmically at every oscillation period. The oscillation period of this compound pendulum is stable for small oscillations (1-2 rad) as in our case. In this way, the position of the head could be estimated and anticipated with precision. This point is now added to the text (lines 138-140).

Q14- Line 112: Which method was used in preliminary experiments to test a possible effect of wind during the swing movement?

R14- In preliminary experiments, 5 volunteers tested the device for this issue and could not perceive any tactile or auditory cue related to the wind. This was briefly stated in the original version (line 147). We have now changed this sentence to provide more details (lines 151-153 of the revised version).

Q15- Line 116: From my understanding of the methods, for each subject, acceleration thresholds were only tested twice. If this was indeed the case, it is not enough to assess the reproducibility of the results, particularly because the aim of this study was to assess this reproducibility.

R15- Similarly to parameters related to sound and oscillation, acceleration was evaluated 4 times (2 test-retest trials). This information was provided at the beginning of this section (lines 87-90 of the original manucript). However, we agree that the sentence “All measurements were repeated twice in a test-retest design.” is confusing. This sentence has been suppressed. For more clarity, the following sentence was added at the beginning of the second paragraph in this section (line 117): A total of 4 tests was designed for each subject and each parameter. After inclusion, subjects underwent a trial of test and retest measuring the latency and the acceleration threshold of movement perception on a swinging bed. A 10-minute interval separated the test and the retest. A second test-retest trial was carried out several days after the first (mean delay between trials 13± 2.1 days, range: 2-50) on the same group. Four subjects were lost to follow-up for the second trial.” 

Q16- Line 116: “all measurements”. Are there many? The authors should specify what these measurements are. 

R16- This sentence has been suppressed. It has been replaced by the detailed information provided above.

Line 119: The methods used to measure the movement perception delay need clarification. 

Q17- Were both forward and backward peaks used for this assessment? 

R17- Only backward peaks were used in order to have only one beep per cycle and allow a larger time lag exploration. This point is now added to the materials and methods (line 163).

Q18- If so, did movement perception differ according to the considered peak?

R18- We did not investigate the difference in perception according to the direction of the movement, but this is certainly a very interesting point.

Q19- How many oscillations were produced by each bed release? How many bed releases were needed for assessing movement perception? Were the oscillations stopped when their amplitude dropped below a given amplitude, before producing a new bed release?

R19- For sound-movement synchronicity, each bed release was followed by 8-10 supra liminary oscillations. Each bed release generally allowed testing 2 time-lags. This point is now added to the materials and methods (lines 171-173).

Q20- The oscillation amplitudes decreased with the number of cycles. Also, why were the subjects asked to estimate the time their head reached their highest position rather than to estimate the time of the downward motion onset? The latter variable would seem more appropriate for estimating movement perception.

R20- We chose the peak because it corresponds to the maximum absolute value of deceleration. The peak also corresponds to a change of direction. Describing it to the patients as the “peak” appeared to be easy to understand for the subjects. This point is now added to the materials and methods.

Q21- Line 152: Does the authors have an explanation for the observed large difference between the sound-peak threshold and the peak-sound threshold?

R21- This is indeed a very interesting point. As shown in figure 2, sound-peak and peak-sound delays are the 2 boundaries of the synchronous perception interval (SPI). Their difference represents the width of SPI and probably the tolerance of the multisensory integration system to time discrepancies between the sensory inputs. This point has now been discussed (lines 283-289).

Q22- Line 185: The sound had to occur before the peak to be considered as synchronous. On the contrary, and as mentioned by the authors, previous studies showed that vestibular sensations were perceived later than sounds. The authors proposed that this could be due to the predictability of the swinging movement in their study, while vestibular stimulation could not be predicted in previous studies. This hypothesis is plausible. Another hypothesis that needs to be considered is that prior to the peak, body acceleration fell below the acceleration threshold for detecting body motion. The authors’ finding could also result from a delay for generating the beep on the basis of bed position signal. It is important to measure and to provide this delay (and as specified above, to specify the methods used to precisely send to beeps prior to the peaks).

R22- Thank you for this remark. This is an interesting hypothesis. However, we should point out that before reaching the peak, the speed decreases but the negative acceleration (deceleration) reaches its maximum absolute value, and what is perceived by the vestibular captors is the negative acceleration and not the speed. In fact, the acceleration curve has a 180° phase-shift with respect to the head position plot. One might argue that this deceleration is perceived differently from the maximum positive acceleration (head at its lowest point) and that PSS may change if the sound is emitted at that point. This idea is now added to the discussion as follows: “Another hypothesis to explain our negative PSS (oscillation peak after the sound perceived as synchronous) is that before reaching the peak, the negative acceleration increases rapidely in its absolute value and this phenomenon may contribute to the inverted temporal relation between sound and movement. It would be interesting to study the effetct of the sound emitted at the point of maximum positive acceleration (head at its lowest point) on the PSS.”

Concerning the sound emission delay, the loudspeaker is placed in the infrared detection box under the subject. The distance between the box and the head is approximately 1.5 m. At the speed of sound, the delay created by this distance is 4 ms (1.6% of the PSS). We have now added the clarification in the materials and methods (lines 132-136).

Q23- Line 210: The authors hypothesized that subjects made temporal corrections based on previous asynchrony. This could be tested by testing whether the asynchrony changed over the experimental session.

R23- Unfortunately, our experimental design does not allow verifying such a hypothesis. We could have proposed a continuous back-and-forth sweeping of the time lag around the point of subjective synchronicity (PSS) and expect a progressive reduction of the synchronous perception interval or a PSS approaching zero with the increasing number of sweeps. This idea is now added as a discussion point (lines: 274-277).

Q24- Line 234: The authors mentioned that the swinging bed stimulated non-vestibular sensory inputs (e.g., somatosensory inputs) before adding “Nevertheless, the measures were reproducible and appeared to be coherent”. What is the link between these two statements?

R24- We apologize for this shortcut in reasoning. The idea was that if there are uncontrolled sensory inputs in this protocol, they might increase the intra-individual and inter individual variabilities of the parameters. We have now clarified the link as follows: “The presence of uncontrolled sensory cues would increase the intraindividual and inter individual variabilities of the parameters, but our measures appeared reproducible and coherent indicating the stability of the sensory cues during the trial.”

Q25- Figure 3 caption. Sound-peak (SP) and peak-sound (PS) thresholds should be defined in the text.

R25- These were defined in the original manuscript (line 126-127). We have now added the abbreviations SP and PS in the revised text.

Q26- Figure 3. The figure shows that 6 subjects had the same P-S threshold, probably -750 ms, i.e. the greatest lag used in the study. I wonder if these data represent the actual subjects’ perception of the synchrony between the sound and the peak.

R26- Yes. We confirm that 750 ms was the actual subjects’ perception of the synchrony.

---

## [Editor Report · Decision Letter 1]

21 May 2021

PONE-D-20-17020R1

Measuring Threshold and Latency of Motion Perception on a Swinging Bed

PLOS ONE

Dear Dr. Bozorg Grayeli,

Thank you for submitting your manuscript to PLOS ONE. After careful consideration, we feel that it has merit but does not fully meet PLOS ONE’s publication criteria as it currently stands. Therefore, we invite you to submit a revised version of the manuscript that addresses the points raised during the review process.

You have satisfactorily addressed all the issues raised by the reviewers.

However, I have one comment and one remark, both minor.

In the surmmary, you should specifiy that the peak is a position one.

In the discussion you should add a comment on the fact that the subject can (or cannot) use information from the beep in the synchronicity task.

Indeed, auditory system is able to detect when the sound source is moving away or approaching using Doppler effect or changes in sound intensity. As velocity at the peack position is zero, the subject knows that the beep is synchronous to the peak position only if he does not perceive the sound source moving. I guess 5ms is too short for a beep to give movement information, but you should comment this (with references).

We look forward to receiving your revised manuscript.

Kind regards,

Pierre Denise, Ph.D, M.D.

Academic Editor

PLOS ONE
---

## [Author Response · Author response to Decision Letter 1]

22 May 2021

Q1- In the summary, you should specify that the peak is a position one.

A1- This information has now been added to the abstract.

Q2- In the discussion you should add a comment on the fact that the subject can (or cannot) use information from the beep in the synchronicity task.

Indeed, auditory system is able to detect when the sound source is moving away or approaching using Doppler effect or changes in sound intensity. As velocity at the peak position is zero, the subject knows that the beep is synchronous to the peak position only if he does not perceive the sound source moving. I guess 5ms is too short for a beep to give movement information, but you should comment this (with references).

A2- We totally agree. This could have been a source of information but changes in the sound spectrum or the doppler effect are minimal for a 5 ms beep (0.16% of the oscillation period). We have now added this discussion together with 3 references as follows (lines 339-343): “Another issue is the detection of the sound source movement relative to the head position by monuaral (spectral changes, doppler effect) [33] or binural functions (interaural time and intensity differences) [34]. These effects are sensitive to signal duration [35]. Considering the shortness of the sound stimuli (5 ms, 0.16% of the oscillation period), this effect can be considered as negligeable.” 

References:

33- Grothe B, Pecka M, McAlpine D. Mechanisms of sound localization in mammals. Physiol Rev. 2010;90:983-1012. doi: 10.1152/physrev.00026.2009.

34- Baumann C, Rogers C, Massen F. Dynamic binaural sound localization based on variations of interaural time delays and system rotations. J Acoust Soc Am. 2015 Aug;138(2):635-50. doi: 10.1121/1.4923448.

35- St George BV, Cone B. Perceptual and Electrophysiological Correlates of Fixed Versus Moving Sound Source Lateralization. J Speech Lang Hear Res. 2020 Sep 15;63(9):3176-3194. doi: 10.1044/2020_JSLHR-19-00289.

---

## [Editor Report · Decision Letter 2]

26 May 2021

Measuring Threshold and Latency of Motion Perception on a Swinging Bed

PONE-D-20-17020R2

Dear Dr. Bozorg Grayeli,

We’re pleased to inform you that your manuscript has been judged scientifically suitable for publication and will be formally accepted for publication once it meets all outstanding technical requirements.

Kind regards,

Pierre Denise, Ph.D, M.D.

Academic Editor

PLOS ONE

Additional Editor Comments (optional):

Note 2 typing mystakes in the paragraph you added (line 341 "monuaral", line 342 "binural")

---

## [Editor Report · Acceptance letter]

29 Jun 2021

PONE-D-20-17020R2 

Measuring Threshold and Latency of Motion Perception on a Swinging Bed 

Dear Dr. Bozorg Grayeli:

I'm pleased to inform you that your manuscript has been deemed suitable for publication in PLOS ONE. Congratulations! Your manuscript is now with our production department. 

Kind regards, 

on behalf of

Pr. Pierre Denise 

Academic Editor

PLOS ONE